# Prevalence of Herpesvirus DNA in Corneal Transplant Recipients

**DOI:** 10.3390/jcm12010289

**Published:** 2022-12-30

**Authors:** Julia Bing Bu, Stephanie D. Grabitz, Norbert Pfeiffer, Joanna Wasielica-Poslednik

**Affiliations:** Department of Ophthalmology, University Medical Center of the Johannes Gutenberg-University Mainz, 55131 Mainz, Germany

**Keywords:** herpetic keratitis, herpes simplex virus 1, varizella-zoster virus, HSV-1-DNA, VZV-DNA, PCR, penetrating keratoplasty

## Abstract

**Purpose:** Graft failure after penetrating keratoplasty (PK) is a serious complication, especially in eyes with herpetic keratitis (HK). This study evaluated the prevalence and graft survival of herpes simplex virus type 1 (HSV-1) and varicella zoster virus (VZV) DNA in recipient corneas during PK. **Methods:** The retrospective study was performed at the Department of Ophthalmology at University Hospital in Mainz, Germany. We analyzed data from every patient who underwent PK between January 2020 and June 2021. According to our clinical routine, we performed HSV-1 and VZV polymerase chain reaction (PCR) on all excised corneal buttons regardless of the primary clinical diagnosis. **Results:** We included 112 eyes of 112 consecutive patients who underwent PK. At the time of PK, 91 (81.25%) patients had no history of HK and 21 (18.75%) patients did. The recipient corneas of 91 patients without a history of HK tested positive for HSV-1 DNA in 12 (13.2%) eyes, for VZV DNA in 3 (3.3%) eyes, and for HSV-1 and VZV DNA simultaneously in 2 (2.2%) eyes. The recipient corneas of 21 patients with a preoperative history of HK tested positive for HSV-1 DNA in 13 (61.9%) eyes and VZV DNA in 1 (4.8%) eye. All patients with positive herpes DNA and no history of HK prior to PK received antiherpetic treatment and had a 100% graft survival rate after 1 year. **Conclusions:** We found herpesvirus DNA in 18.7% of recipient corneas without clinical suspicion or history of herpes keratitis. This suggests the need of routine HSV-1 and VZV PCR testing in all explanted corneas regardless of clinical suspicion, to detect, treat and prevent possible recurrence of herpes infection in corneal grafts and support graft survival.

## 1. Introduction

Herpetic keratitis (HK) is one of the major causes of infectious corneal vision impairment [1]. Herpes simplex virus type 1 (HSV-1) and varicella-zoster virus (VZV) are the main herpesviruses that cause herpetic keratitis [2]. The majority of the world’s population will become infected with HSV-1 and VZV at some point in their lives. The prevalence increases with age [3]. Ocular manifestations can occur upon initial exposure to the virus or in association with latent infection. The latency in nervous tissue caused by HSV-1 and VZV is an intriguing feature of herpes viruses [4,5]. The clinical manifestation of HK can range from epithelial keratitis, stromal keratitis, endotheliitis to metaherpetic keratitis [6].

Penetrating keratoplasty (PK) is one of the oldest and most successful human tissue transplants due to the immune privileged status of the cornea. Despite the increasing popularity of lamellar keratoplasties, PK remains an established procedure for various indications, including ulcers (noninfectious/viral/bacterial), corneal degeneration and dystrophy, scarring, stromal opacities, chemical or thermal injury, keratoconus, mechanical trauma, or regraft [7]. Graft failure is a serious complication, especially in HK eyes, although the incidence of immune graft rejection has decreased significantly in recent decades thanks to improvements in the preoperative and postoperative management, surgical techniques, effective immunosuppressive medications, and advanced eye banking standards [8]. Surgical trauma to the corneal nerve plexus and modulation of the ocular immunologic response caused by postoperative steroid treatment can affect surgical outcomes, particularly in HK eyes. PK in patients with a history of HK have a higher rate of postoperative complications compared to patients with non-herpetic diseases. Complications include recurrent HK with persistent epithelial defects, inflammation, vascularization, scarring, and ultimately graft failure [9,10,11]. Due to reactivation of the herpesvirus in the sensory ganglion, from where it can migrate into the eye via the nerve axon, recurrent HSV-1 and VZV infections can occur after PK [12]. To reduce the risk of HK recurrence in the graft, adequate topical and especially systemic antiherpetic treatment is required in addition to the immunosuppressive therapy with local steroid eye drops [13,14,15]. Therefore, it is of great importance to identify patients who are at risk for developing HK after PK.

The purpose of this study was to determine the prevalence of HSV-1 and VZV DNA in excised corneas from a large cohort of consecutive patients who underwent PK. We aimed to find out how often the HSV-1 or VZV polymerase chain reaction (PCR) was positive even though the patient had no history of HK and to evaluate their graft survival.

## 2. Material and Methods

We conducted a single-center retrospective study at the Department of Ophthalmology of the University Medical Center Johannes Gutenberg University in Mainz, Germany. We included all consecutive patients who underwent PK between January 2020 and June 2021. According to our clinical routine, we performed HSV-1 and VZV PCR assays on all excised corneal buttons, regardless of the primary clinical diagnosis. The PCR assays were performed at the Department of Virology, University Medical Center of Johannes Gutenberg University in Mainz, Germany. The following patient data were collected: demographic data, ocular history (especially regarding previous ocular herpes infections), uncorrected visual acuity, results of preoperative HSV-1 and VZV PCR swabs (if available), and results of HSV-1 and VZV PCR tests of all excised corneal buttons. Preoperative history of HK was obtained based on typical clinical findings and/or history of positive HSV-1 or VZV PCR swabs. All patients with a history of HK received around 4 weeks preoperative prophylactic topical and perioral antiviral treatment (acyclovir/ganciclovir ophthalmic ointment 1–5 times daily, acyclovir 5 × 400–800 mg/valacyclovir 2 × 500–1000 mg). Postoperative, all patients received topical and perioral antiviral treatment with a preoperative history of HK or with positive HSV-1 or VZV DNA PCR results in corneal biopsies (acyclovir/ganciclovir ophthalmic ointment 1–5 times daily, acyclovir 5 × 400 mg for the first 4 weeks and later 2 × 400 mg/valacyclovir 2–4 × 500 mg for 1 year).

Our standard treatment after PK includes dexamethasone 1.3 mg/mL or prednisolone acetat 10.5 mg/mL eye drops 6 times daily with monthly reduction to one maintenance dose per day for a total of at least 1 year; antibiotic treatment with, e.g., ofloxacin 3 mg/dL eye drops without preservative 4 times daily until epithelial closure, and preservative-free artificial tears 6 times daily.

## 3. Results

We analyzed 112 eyes of 112 consecutive patients, who received PK and herpes DNA PCR testing of the excised corneal buttons. At the time of PK, 91 (81.25%) patients had no history of HK and 21 (18.75%) patients did. Primary diagnosis for PK included acanthamoeba keratitis, trauma, corneal ulcer, corneal decompensation, corneal dystrophy, herpetic keratitis, keratoconus and neurotrophic keratopathy. Forty-two (37.5%) patients had a history of previous PK and 11 (9.8%) had a history of DMEK or DSAEK. 35.1% of the cases in the group without HK were a repeat-to-intial keratoplasty compared to 47.6% of the cases in the group with HK.

The recipient corneas of 91 patients without a history of HK were tested positive for HSV-1 DNA in 12 (13.2%) eyes, for VZV DNA in 3 (3.3%) eyes, and for HSV-1 and VZV DNA simultaneously in 2 (2.2%) eyes. The recipient corneas of 21 patients with a preoperative history of HK were tested positive for HSV-1 DNA in 13 (61.9%) eyes and for VZV DNA in 1 (4.8%) eye. Graft survival was observed for a follow-up period of 1 year. All patients with a positive herpes DNA in the excised corneas and no history of HK prior to PK received antiherpetic treatment and had a 100% graft survival rate after 1 year.

Additionally, 45 (40.2%) patients had corneal swab results of HSV-1, VZV and CMV DNA from their past due to different indications (e.g., conjuncitivitis, keratitis, corneal ulcer). As this is not a standard service performance prior to PK, herpesvirus PCR swab tests are not routinely performed in patients with for example keratoconus, bulbar trauma, burns, or corneal dystrophy.

In both groups, preoperative visual acuity was logMar 1.9 (1.3, 2.3). Visual acuity improved 1 year postoperative to logMar 0.7 (0.3, 1.6) in the group without a history of HK and to 1.0 (0.7, 1.3) in the group with a history of HK.

All results can be found in Table 1.

## 4. Discussion

In this study, we determined the HSV-1 and VZV PCR rates of corneal buttons in 112 patients, who received PK during a timeframe of 18 months between January 2020 and June 2021. 66.7% of patients with a history of HK had a corresponding positive herpes DNA result in the corneal biopsy. This result is within the range of those reported by Remejer et al. [2], Jeng Y-T et al. [16], and Shimomura et al. [12], who detected HSV-1 DNA in the recipient corneas of patients with a history of ocular herpes in 48%, 63.3% and 85.7% of cases, respectively. A history of HK doesn’t automatically lead to a positive corneal biopsy result. Several factors can play an influencing role here. It is less likely to detect viral DNA in corneas, the longer the duration following the active episode, most likely due to the decline in the amount of viral DNA [17,18]. In addition, every patient with a history of HK gets an antiviral topical and systemic treatment at least 4 weeks prior to PK, which can influence results of herpes DNA PCR testing due to inhomogeneous DNA distribution or low herpes virus DNA load. Moreover, this group consists of a repeat-to-initial keratoplasty in 47.6%, confirming a higher graft failure rate in HK eyes. This fact can also have an impact on the corresponding herpes DNA results.

In the group of patients with no history of HK, positive herpes DNA was found in 18.7% with a primary clinical diagnosis non-related to herpes (e.g., trauma, NK, lagophthalmus, corneal dystrophy, keratoconus). In contrast, Seitz et al. found a comparable low prevalence of only 1.5% HSV PCR DNA in HK negative recipient corneal samples [19]. Considering the fact, that a history of HK doesn’t show a corresponding positive herpes DNA result in biopsy, also raises the question of false negative biopsy results in the group with no history of HK. For example, an unnoticed or detected herpes infection from the past with a present low virus DNA load cannot be detected by PCR testing and may lead to a false negative biopsy.

A general higher prevalence of HSV-1 DNA than VZV DNA in corneal tissue was expected because of a relative low recurrence rate of VZV compared to HSV-1 in immunocompetent patients [2,20] The herpes PCR test results of the swab and biopsy were congruent in 73.3%. On the other hand, 11.1% had a positive herpes DNA swab and a negative herpes DNA biopsy result, and 17.8% had a negative herpes DNA swab but a positive herpes DNA biopsy result. Therefore, the herpes DNA biopsy and swab results should be considered complementary, if available.

Forty eight percent of patients with a history of HK had at least one keratoplasty in the past. With begin of preoperative antiviral treatment lasting the postoperative period of one year, 90.5% had no recurrence or graft failure within the follow-up of 1 year. Among patients without history of HK, 35.0% had at least one keratoplasty in the past. Of 32 patients without a history of HK who underwent repeat-to-initial keratoplasty, 5 patients (16%) had a positive herpes DNA PCR result in biopsy. All patients without history of HK but with a positive corneal biopsy for HSV and/or VZV received postoperative antiherpetic treatment. There was no graft failure within the follow-up of 1 year in this group. This underlines the benefit of standardized corneal biopsy in all PK patients regardless their history of HK and shows a positive postoperative outcome of antiherpetic treatment on graft survival. Taking the positive postoperative outcome of antiherpetic treatment, the high prevalence of herpes infection in the population and the likelihood of an association between known or unknown HK and repeat PK into consideration, one can discuss to implement preventive postoperative antiviral treatment under risk benefit consideration for every repeat-to-initial PK. One could argue that systemic valacyclovir or acyclovir treatment is overall well tolerated even for long term use and resistance development is rare [21]. However, we are not in favor of a recommendation for preventive antiherpetic treatment with lack of evidence. Antiviral treatment should be reserved for those with solid indication. Preventive antiviral treatment for all repeat-to-initial PK or all PK in general limits the diagnostic value of herpes DNA PCR testing in biopsy and raises questions about the duration and intensity of this preventive treatment and certainly increase the relatively small risk of side effects or resistance.

Visual acuity without correction showed an increase in the follow-up period of 3 months and 1 year. Since the corneal sutures after PK were removed after 1 year at the earliest, no best corrected visual acuity was obtained for irregular astigmatism. Usually we propose 6–8 weeks after removal of the corneal sutures corresponding refraction correction.

Our findings suggest that the prevalence of herpes DNA in corneal tissue of the population is higher than expected and this should be considered for perioperative PK treatment. Remeijer et al. have shown that preoperative antiherpetic treatment influenced the HSV-1 DNA load in corneas and that patients with a clinical history of HK tend to have a higher risk of graft failure. An inverse correlation between intracorneal HSV-1 DNA load and graft survival in patients with HK had been demonstrated [2]. Additionally, it was shown that the incidence of HSV epithelial keratitis after corneal transplantation was six times higher than in the general population, even if there was no history of HK [22,23]. VZV DNA was found in corneal buttons from patients with herpes zoster ophthalmicus (HZO) up to 61 years after the onset of HZO [24,25,26].

To date, herpes DNA PCR screening of the recipient corneas undergoing PK is not established. The same applies to the donor corneas. Remeijer et al. screened a large cohort of 273 donor corneoscleral rims after PKP for the presence of herpes DNA. No HSV-2 or VZV DNA was detected, and only 2 (0.007%) of 273 corneoscleral rims showed HSV-1 DNA. More interestingly, both HSV-1 DNA positive donor corneas were transplanted in patients with a positive history of herpes keratitis and they didn’t develop post-PK HK recurrence or graft failure for a follow-up more than 4 years [2]. In a recent study, Qu et al. found HSV-1 DNA in 0.7% (7/942) donor corneas and 0.6% (6/942) VZV-DNA [27]. They, and some authors, have reported that HSV-1 reactivation after PK can occur through donor-to-host transmission [27,28,29,30]. Therefore, donor graft selection with respect to herpes DNA may also be required and should be subject of further investigation.

For ocular herpetic infection, topical antiviral medication alone is not sufficient. Systemic antiherpetic treatment is needed to prevent recurrence and graft failure after PK [23]. Both acyclovir and valacyclovir are effective as systemic antiherpetic agents [23]. In cases when no antiviral prophylaxis was used postoperatively, the risk of HK recurrence was raised up to 55.0% and the risk for graft rejection up to 52.0% [13,31,32]. If there is a known ocular history of HK or sufficient evidence by herpes PCR swab or biopsy test, we recommend timely initiation of antiviral prophylaxis in preparation before or at the latest after PK. The efficacy of topical and especially systemic antiherpetic treatment in combination with topical steroids and its impact on reducing the complication rate, herpetic recurrence and graft survival is indisputable [23,33].

## 5. Conclusions

We found herpesvirus DNA in 18.7% of recipient corneas without clinical suspicion or history of HK. These results demonstrate the importance of HSV-1 and VZV DNA PCR screening of the recipient corneas of all patients undergoing PK with and without a known history of HK to detect, treat and prevent possible recurrence of herpes infection in corneal grafts and support graft survival. Most of the world’s population will become infected with HSV-1 and VZV during their lifetime. Patients with no clinical signs or history of HK and an undetected primary infection are at risk for complications after PK, either through reactivation due to neuronal or corneal latency or through transmission from corneal grafts [29,34]. Standardized corneal biopsy for herpes virus can prevent herpes associated complications and graft failure, especially in the group of patients without a history of HK, who are least expected to have herpes DNA.

## Figures and Tables

**Table 1 jcm-12-00289-t001:** Results of HSV-1 and VZV DNA in corneal transplant recipients for PK.

	No History of HK		History of HK	
Number	**91**		**21**	
Age (years)	59.62 ± 21.21		65.33 ± 14.41	
Gender (female)	31 (34%)		11 (52%)	
Repeat-to-initial keratoplasty	32 (35%)		10 (48%)	

Herpes DNA-PCR of recipient corneas	**−**	**+**	**−**	**+**
HSV-1	79 (87%)	**12 (13%)**	8 (38%)	13 (62%)
VZV	88 (97%)	**3 (3%)**	20 (95%)	1 (5%)
HSV-1 & VZV	89 (98%)	**2 (2%)**	21 (100%)	0 (-)
Repeat-to-initial keratoplasty	27 (30%)	**5 (5%)**	6 (29%)	4 (20%)
Primary diagnosis for PK				
Acantamoeba keratitis	9 (10%)	0	0	1 (5%)
Trauma	10 (11%)	4 (4%)	1 (5%)	0
Corneal ulcer	18 (20%)	7 (8%)	2 (10%)	1 (5%)
Corneal decompensation	9 (10%)	0	0	0
Corneal dystrophy	7 (8%)	1 (1%)	0	0
Neurotrophic keratopathy	4 4%)	2 (2%)	0	0
Keratokonus	17 (25%)	3 (3%)	0	1 (5%)
Herpetic keratitis	0	0	4 (19%)	11 (52%)
preoperative antiviral therapy	-	-	7 (100%)	14 (100%)
postoperative antiviral therapy	-	17 (100%)	7 (100%)	14 (100%)
**Graft survival 1 year follow-up**	69 (93%)	**17 (100%)**	6 (86%)	13 (93%)
**Visual acuity (logMar) without correction**				
preoperative	1.90 (1.3, 2.3)		1.90 (1.3, 2.3)	
3-month follow up	0.90 (0.4, 1.6)		1.20 (0.7, 1.9)	
12-month follow up	0.7 (0.3, 1.6)		1.0 (0.7, 1.3)	

## Data Availability

Data are available on request.

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
