# Peer review of "Prevalence of Herpesvirus DNA in Corneal Transplant Recipients"

_jcm, 2022, doi:10.3390/jcm12010289_

Round 1
Reviewer 1 Report
The problem of studying the prevalence and graft survival of herpes simplex virus type 1 and varicella zoster virus DNA in recipient corneas during PK remains relevant until now. The manuscript is clear, relevant for the field and presented in a well-structured manner. Unfortunately, 75% of the cited references aren’t the recent publications (within the last 5 years), but, nevertheless, they remain relevant. The table properly show the data obtained during the study. The conclusions are consistent with the evidence and arguments presented. The authors' conclusions that a standardized corneal biopsy for the presence of herpes virus can prevent herpes-related complications and graft rejection, especially in the group of patients without a history of HK, are reasonable and logical. But, I want to know if a PCR analysis was performed for the presence of viruses in the donor material?
Reviewer 2 Report
This is a well-written manuscript of a stringent study. My comments and suggestions are:
1. The strength of the study is a large consecutive number of cornea buttons are examined uniformly. In addition, the uniform treatment of the patients pre- and post- PK further increases the quality of the study.
2. Although this is a frequently studied topic, I think the data adds to our understanding in this field and may contribute to future practice pattern change.
3. Data presentation improvement suggestions: Table 1
- Add number to each of "Herpes DNA-PCR of recipient corneas" next to "-"/"+". If you are including HSV and VZV both, the numbers will help understanding assuming all the percentage data in the columns below is based on the total number of that column.
- The "HSV-positive preoperative swab" line does not really help with your main idea. It is interesting to see that discrepancy but maybe just mention it in result and discussion without showing it in the table
4. It would be interesting to see the breakdown of the 35% of patients without HK history but had more than one PK in the past (line 140). What percentage of those buttons got a positive PCR? Same for the patients with a positive HK history. Adding this line to the table would be informative.
5. The fact that not all PCR was positive in patients with HK history raise the question of false negative in the "no history of HK" population. It should be mentioned.
6. Should all repeat PK recipients just get anti-viral treatment? Based on the answer to #4 and concern of #5, the author can address the question.
7. Given the high prevalence of Herpes infection in the population as the authors pointed out, should anti-viral regimen be applied to all PK recipients? What's the cost of testing and what's the cost of standard treatment? It is interesting to think about the practicality.
